# Efficiency of Simulation-Based Learning Using an ABC POCUS Protocol on a High-Fidelity Simulator

**DOI:** 10.3390/diagnostics14020173

**Published:** 2024-01-12

**Authors:** Robert Simon, Cristina Petrisor, Constantin Bodolea, Adela Golea, Sara Hora Gomes, Oana Antal, Horațiu Nicolae Vasian, Orlanda Moldovan, Cosmin Ion Puia

**Affiliations:** 1Doctoral School, Faculty of Medicine, University of Oradea, 410087 Oradea, Romania; cosmin.puia@umfcluj.ro; 2Anesthesia and Intensive Care Department, “Iuliu Hatieganu” University of Medicine and Pharmacy, 400347 Cluj-Napoca, Romania; petrisor.cristina@umfcluj.ro (C.P.); horatiumd@gmail.com (H.N.V.); 3Clinical Institute of Urology and Renal Transplant, 400000 Cluj-Napoca, Romania; 4Clinical County Emergency Hospital, 400347 Cluj-Napoca, Romania; adeg2810@gmail.com; 5Municipal Clinical Hospital, 400139 Cluj-Napoca, Romania; 6Emergency Medicine Department, “Iuliu Hatieganu” University of Medicine and Pharmacy, 400347 Cluj-Napoca, Romania; 7Life and Health Sciences Research Institute (ICVS), School of Medicine, University of Minho, Campus Gualtar, 4710-057 Braga, Portugal; 8ICVS/3B’s-PT Government Associate Laboratory, 4710-057 Braga, Portugal; 9Regional Institute of Gastroenterology and Hepatology, 400394 Cluj-Napoca, Romania; 10Clinical Emergency Hospital for Children, 400177 Cluj-Napoca, Romania; 11Surgery Department, “Iuliu Hatieganu” University of Medicine and Pharmacy, 400347 Cluj-Napoca, Romania

**Keywords:** point of care ultrasound, airway ultrasound, lung ultrasound, heart ultrasound, simulation, simulation based medical education, critical patient ultrasound, ultrasound protocol

## Abstract

Critically ill patients with rapidly deteriorating clinical status secondary to respiratory and cardio-vascular compromise are at risk for immediate collapse if the underlying pathology is not recognized and treated. Rapid diagnosis is of utmost importance regardless of the setting. Although there are data to support the use of point-of-care ultrasound in critical patients, there is no consensus about the best educational strategy to implement. We designed a curriculum based on the ABC (Airway, Breathing, Circulation) protocol that covers essential airway, lung, and cardiac ultrasound skills needed for fast diagnosis in critical patients and applied it in high-fidelity simulation-based medical education sessions for anesthesia and intensive care residents year one and two. After theoretical and practical assessments, our results show statistical differences in the theoretical knowledge and above-average results in practical assessment. Our proposed curriculum based on a simple ABC POCUS protocol, with an Airway, Breathing, and Circulation approach, is useful in teaching ultrasound basics regarding airway, lung, and cardiac examination using high-fidelity simulation training to anesthesia and intensive care residents, but further research is needed to establish the utility of Simulation-Based Medical Education in Point of Care Ultrasound in the critical patient.

## 1. Introduction

Critically ill patients with rapidly deteriorating clinical status secondary to respiratory and cardio-vascular compromise are at risk for immediate collapse if the underlying pathology is not rapidly recognized and treated. In these cases, a rapid diagnosis is of utmost importance regardless of whether the patient is in an intraoperative, perioperative, or intensive care unit (ICU) setting. Many of the etiologic causes for respiratory insufficiency and shock can be diagnosed at the bedside using dynamic point-of-care ultrasound examination. However, not all anesthesiologists and intensivists have ultrasound skills, and the variability in the depth of knowledge might be substantial. However, the current management of such demanding patients requires remarkable knowledge and skills. Point of Care Ultrasound (POCUS) has gained a lot of attention in recent years, especially in the course of the recent pandemic. Nowadays, POCUS is an essential tool in most critical care departments and emergency departments, with the latest developments in its use for anesthesia and perioperative medicine [1,2,3,4,5]. There are recommendations from the European Society of Intensive Care Medicine regarding the basic ultrasound skills required by doctors working in the ICU [1]. We have designed an ABC algorithm containing essential ultrasound notions for the rapid diagnosis of patients with acute decompensation of respiratory and hemodynamic parameters during the pandemic [6].

Although there are data to support the use of POCUS in the intensive care unit and in critical patients, there is no consensus about the best educational strategy to be implemented. Currently, there is a lack of data regarding the best way to promote physician skills acquisition to properly perform ultrasound images, interpret US scans, and integrate the information from the ultrasound examination with the clinical scenario in anesthesia and critical care clinical reasoning algorithms. Similarly, the formal inclusion of ultrasound examinations in standardized curricula in anesthesia and intensive care medicine residency programs is not yet generally accepted in different countries.

Accurate, correct diagnostic methods and medical education are prerequisites for rapid treatment of life-threatening events during surgery and during the ICU stay. From this point of view, it would be reasonable to suggest that the acquisition of basic ultrasound skills by anesthesia trainees using high-fidelity simulation training could improve the outcome of their future patients. However, up to now, there is scarce data on the impact of simulation training on the performance of anesthesia and intensive care performance regarding ultrasound diagnosis.

Since simulation-based medical education (SBME) is currently used in other aspects regarding anesthesia and intensive care practice, like airway management, complex scenarios regarding anesthesia crisis, trauma, pediatric intensive care, and ultrasound could be learned in the simulation laboratory since high fidelity mannequins with large ultrasound image libraries are currently available. Recently, the European Society of Anesthesia and Intensive Care recommended the integration of simulation-based medical education (SBME) in the curriculum of specialist training, and POCUS stands in the top 10 of the procedures that residents should be able to learn using simulation [7]. Even so, SBME varies widely around Europe, with only a few countries having a well-implemented system [8]. It is suggested that ultrasound skills are needed for anesthesia and intensive care residents, and simulation training is recommended [8].

However, there is a lack of data regarding the process of learning POCUS via medical simulation using high-fidelity mannequins [9]. Even more importantly, ultrasound knowledge and skills are required, and finally, the level of ultrasound expertise is not yet defined.

Thus, the aim of this prospective interventional study is to implement a straightforward ABC algorithm using point-of-care ultrasound, which is designed for the diagnosis of acute reversible causes for breathing and cardio-vascular insufficiency in a curriculum for anesthesia and intensive care trainees with no previous expertise in ultrasound examinations and to evaluate the efficiency of education using high-fidelity ultrasound simulation.

## 2. Materials and Methods

### 2.1. Ethical Approval

After designing the curriculum and the simulation program, the ethics commission from the University of Medicine and Pharmacy from Cluj-Napoca approved the study (approval document nr AVZ75/14.03.2022). An invitation to participate in the simulation program was sent to all anesthesia and intensive care residents from the first and second year of training from Cluj-Napoca. Participants were notified about the possibility of participating in the current study. Oral and written informed consent was obtained from all participants.

### 2.2. Study Design

We designed a curriculum based on the ABC (Airway, Breathing, Circulation) protocol that covers essential airway, lung, and cardiac ultrasound skills needed for fast diagnosis in critical patients [6]. In this curriculum, the airway assessment (point A of the protocol) consists of the identification of the correct placement of the endotracheal tube using ultrasound. Lung ultrasound (point B of the protocol) consists of a diagnosis of life-threatening diagnosis: pneumothorax, large pleural effusion, interstitial syndrome, and lung consolidation. The curriculum is based on well-established data from the literature [2,4,10]. Cardiac transthoracic and inferior vena cava ultrasound (point C of the protocol) focuses on the acquisition of the following ultrasound windows: substernal four chambers, apical four chambers, parasternal long and short axis, and inferior vena cava ultrasound, with recognition of the important structures from these views [3]. The diagnosis that the curriculum focuses on is cardiac tamponade, acute right heart failure, severe depression of the left ventricle, and assessment of inferior vena cava.

Participants were notified about the possibility of participating in the current study. The inclusion criteria are anesthesia and intensive care residents from Cluj-Napoca years one and two, being able to take the pre-course MCQ test, being able to participate in all 3-simulation sessions, and being able to participate in the last evaluation session. The exclusion criteria are having performed an ultrasound or POCUS course that focuses on airway, lung, or heart ultrasound, not being able to participate in all three continuous simulation sessions and the final evaluation session. Residents who met the inclusion criteria were asked to consent to participate in the study and take a pre-study test in Google Forms. The program for the simulation sessions was made by the coordinator of our simulation laboratory (SIMLAB) from the Center of Practical Abilities at the University of Medicine and Pharmacy “I. Hatieganu” from Cluj-Napoca, where the simulation sessions took place.

The simulation program was divided into three sessions, each being two hours long. All sessions included a brief theoretical presentation using a Microsoft PowerPoint presentation support regarding the objectives of the session that lasted thirty minutes and the main subjects of the session, and a practical part being held on a high-fidelity ultrasound mannequin BodyWorks Eve, which lasted ninety minutes. The practical examination did not include the airway ultrasound because the mannequin did not support this function. The structure of the sessions was as follows: the first session included notions of airway ultrasound and lung ultrasound, the second session was dedicated to heart ultrasound, and the third and last session was dedicated to applying the whole protocol in clinical case scenarios. The groups consisted of a number maximum of 10 residents.

After finishing the simulation program, they were needed to take another MCQ test and a 5-min practical evaluation that was filmed. The video included the mannequin, the probes, the screen of the simulator, and the hands of the resident doing the scan. For the pre-course and post-course MCQ tests, we established an arbitrary value of 70 for a successful pass.

The cases for the evaluation were randomly selected from the main diagnosis presented in the curriculum from lung ultrasound and cardiac transthoracic ultrasound. We were not able to practice and evaluate the airway ultrasound because the mannequin did not support this function. The videos were edited to remove sound and were uploaded to Google Drive with restricted access. The videos were examined by three examiners with experience in ultrasound and POCUS in critical patients in the intensive care unit or emergency unit setting. The evaluation criteria for the practical examination can be seen in Appendix A, which can be found in the Appendix A.

To grade every criterion, the examiners used a Global Rating Scale of overall Performance from 1 to 5. (1—very poor, 2—poor, 3—average, 4—good, 5—very good).

### 2.3. Statistical Analysis

Data were analyzed for distribution, followed by appropriate statistical analysis to test for differences after the intervention was performed. Numerical values were expressed as mean (+/− standard deviation). Paired Samples *t*-test was performed to test if the intervention had any effect on the group.

To assess the rater agreement for the practical examination score, Cohen’s Weighted Kappa was performed.

Interpretation of Cohen’s weighted kappa can be as follows:Less than 0: poor agreementBetween 0.01 and 0.20: slight agreementBetween 0.21 and 0.40: fair agreementBetween 0.41 and 0.60: moderate agreementBetween 0.61 and 0.80: substantial agreementBetween 0.81 and 1: Almost perfect agreement

## 3. Results

Although fifty-five residents attended the simulation sessions, only thirty-eight took part in the final assessment session and were recorded and added to the final analysis, as seen in Figure 1. Sixteen residents were in the first year, and twenty-two were in the second year of training.

MCQ test score results for the pre-course test and the post-course test are highlighted in Figure 2.

As highlighted in Figure 2, for the pre-course MCQ test, 23.69% of candidates fell short of obtaining a 70 score for the test. The results of the post-course test show that 100% of participants obtained a score >70, and a larger percentage of participants obtained scores between 80–89 and 90–100 in the post-course test (47.36% vs. 42.1% for the 80–89 category and 42.1% vs. 7.89% in the 90–100 category).

Descriptive data regarding the MCQ result scores for the pre-course and post-course tests can be seen in Table 1. Statistical analysis using Paired sample *t*-test shows a significant statistical difference (*p* < 0.001, Table 2) after the intervention was applied to the group. The results are highlighted in Table 2.

The percentage of correct final diagnoses for the practical examination can be seen in Figure 3.

The cases for the practical exam were randomly selected from the diagnosis included in the curriculum. The diagnosis was assigned a number, and then, using a random number generator, we allocated a diagnosis for the resident that followed for the examination. The diagnosis selected for the practical examination and the overall frequency of each diagnosis can be seen in Figure 4. Eighty-four percent (84.21%) of the participants succeeded in establishing a correct ultrasonographic diagnosis.

The overall results of the practical examination can be seen in Figure 5.

A full overview of the scores obtained for each evaluation criteria, overall practical examination score, pre and post-course MCQ test results, as well as the diagnosis for the practical examination, can be seen in Appendix A, which can be found in the Appendix A. The overall practical examination scores show that 63.15% of participants were rated in the 4–5 score category (good to very good), 34.21% fall in the 3–4 score category (average to good), and only 2.64% of participants performed below an overall score of 3. The inter-rater agreement for the overall practical examination score (average kappa 0.471, Table 3) showed a moderate agreement. Also, even though establishing a correct ultrasonographic diagnosis was just one of the criteria in the evaluation, 84.21% of the participants succeeded in doing so.

Average Cohen’s weighted kappa for all criteria of evaluation can be seen in Table 4. For the practical examination final score, we calculated Cohen’s Weighted Kappa to evaluate the inter-rater agreement and an average kappa to estimate the overall agreement. These results can be seen in Table 3.

## 4. Discussion

POCUS is used on a large scale in the critical care setting. There are recommendations and enough data in the literature to support its use in the intensive care setting [1,2,3,4]. The way POCUS is used in the intensive care setting varies among centers. This might be because teaching POCUS to residents varies largely among centers due to a lack of standardization in the training curricula regarding POCUS and its educational strategies.

Classical teaching methods, “see one, do one, teach one”, have been put to the test. Concerns regarding patient safety, reduction in working hours for resident doctors, and having the “chance” to see or perform a specific procedure have led medical educators to turn to simulation-based medical education [11]. SBME focuses on recreating real medical scenarios in a safe environment for both patients, who are not exposed to doctors inexperienced in some procedures, and trainees who benefit from the time allocated to learning in a safe psychological climate where there is space for error. This will give the trainee the opportunity to become familiar with uncommon scenarios or things they may rarely encounter in their clinical practice. SBME gives the opportunity for new skill acquisition and retention, as well as increased confidence and development of non-technical skills while increasing patient safety [9,11,12].

When talking about SBME regarding POCUS, we must take into consideration that there is no set curriculum for POCUS training in critical care medicine or anesthesiology. There is some data in the literature regarding the usefulness of teaching POCUS in critical care patients using SBME [9,11,13,14]. There are clear benefits of SBME that translate to increased favorable clinical outcomes, increased self-confidence, and reduced errors [15]. The transition from simulation to clinical practice is another important step in medical education. Although competency is an important factor when talking about medical skills, it is hard to define competency in POCUS. There are data in the literature regarding the clinical outcomes in the scenario where SBME was performed that may suggest simulation intervention is superior to no interventions and to non-simulation training at all [16]. There are data in the literature that validates the transition from SBME for specific procedures from the simulation laboratory to clinical practice [17].

A recent study that tried to define and address the needs of anesthesiology in SBME concluded that airway and ultrasound sessions were the most prominently used and desired topics. The same study addresses some limitations regarding SBME. Only 16% of responders have a specialized department and simulation laboratory dedicated to anesthesia. Although the importance of SMBE is highlighted, most of the responders affirm that they benefit from 1–4 sessions of simulations a year. Some drawbacks are explained by lack of facility, lack of time for the sessions to occur, and lack of trainers. The existence of specialized anesthesia simulation departments with specific personnel correlates better with attendance [15].

Although it is widely believed that the volume of exposure to examinations may lead to better competence in the use of POCUS, regarding simulation-based medical education for this precise topic, some studies demonstrated that it might be a matter of competency rather than the number of examinations [18]. Although the recommendations for basic ultrasound skills to intensivists [1], the definition and assessment of skills proficiency is difficult. A lot of work has been published on this topic, mainly trying to define competency, standardize assessment of competency, and use SBME to obtain this competence [19]. The joint consensus statement from The European Section/Board of Anesthesiology and European Society of Anesthesia and Intensive Care from 2020 marks the importance of SBME for Competency-Based Medical Education and Training. At the same time, one problem regarding SBME stands in assessing the effects of the simulation on clinical outcomes. This is especially relevant for rare clinical situations in which trainees didn’t have the chance to be exposed nor to develop the needed competency during their training. Simulation may be the controlled exposure needed for competency. However, there is insufficient evidence to support the validity of an assessment in a simulation setting when compared with clinical practice [20]. Traditionally, the number of procedures performed by a trainee was essential when talking about competence. Although there are data that supports that the number of procedures performed and year of residency training correlates with successful clinical procedures, that may not be the case when talking about competency, and the number of procedures or years of experience are not correlated with overall performance [21].

There are many proposed protocols for POCUS in critical patients, many of them with demonstrated clinical relevance. Such protocols used currently in clinical practice are the BLUE and FALLS protocol [10], the eFAST protocol for trauma, the eFATE protocol, and the SESAME protocol for cardiac arrest [22]. Our proposed airway, breathing, and circulation protocol [6] encompass the basic ultrasound diagnosis techniques proposed by these protocols focused on life-threatening situations, structured as the ABC approach, and can be applied in different critical case scenarios (ex., deteriorating patient on the clinical ward, critical patient in the emergency department or in the intensive care unit. etc.) The particularity of this protocol is the fact that it can be applied in a structured and oriented approach focused on severe life-threatening situations that allows the integration and correlation of the cardio-pulmonary characteristics of the critical patient.

The advantages of our ABC POCUS protocol are that it is easy to apply and requires a few ultrasound windows to be scanned in a systematic manner, which allows the identification of most of the causes of breathing and hemodynamic deterioration. As taught in our designed curriculum, the acquisition of this limited US knowledge is possible in only a few sessions.

In our study, we found significant statistical differences before and after the intervention regarding the theoretical knowledge of the participants. These findings suggest that the simulation sessions improved the theoretical knowledge of the participants and increased the number of participants who successfully passed the MCQ test from 76.29% to 100%.

More than half of the participants fall into the 4–5 score category for the practical examination, 97.38% fall above three overall scores in the practical examination, and 84.21% of the participants established the correct diagnosis combined with an average inter-rater agreement for the final practical score that rates “fair agreement”, lead us to believe that the curriculum and protocol can be applied in a simulation-based setting. Our described curriculum can be used to teach residents the basics of airway, lung, and cardiac ultrasound in the evaluation of a critical patient. High-fidelity ultrasound simulation training improves both knowledge and skills, as demonstrated by our data.

Regarding the inter-rater agreement on the evaluated criteria, the most substantial agreement was on criteria number 5 and 6, interpretation of images, and documentation of examination. Criteria 3 and 4, systemic examination of the lungs and heart only obtained a fair level of agreement. In criteria 1, 2, and 7, applied knowledge of the equipment, image optimization, and duration of evaluation, we recorded only slight inter-rater agreement. This variability between inter-rater agreements can be explained by the fact that ultrasonography is a user-dependent technique, and so a level of inter-rater disagreement can be seen as demonstrated in the literature [23]. Similar conclusions have been presented by others. A 3-year evaluation of a POCUS training course for physicians in Japan concluded that specific interventions could improve the POCUS knowledge among participants. Although the difference in the pre-intervention evaluation scores existed and was correlated with more years of experience, clinical rank, and frequency of POCUS use, the post-intervention differences among participants were statistically insignificant [24].

There are some limitations to our single-center study. First, it is hard to evaluate the usefulness of a simulation-based curriculum and the applications it has in clinical practice, and we did not aim to examine this endpoint. The transition from competency on the high-fidelity mannequin to clinical practice is another important aspect of medical education. Further studies are needed to determine the real implications of SBME regarding POCUS and its effects on clinical outcomes. There are data to support the usefulness of SBME for POCUS in acquiring basic knowledge, but the transition to clinical practice may imply further continuation of medical education. SBME may not be enough for transition into unsupervised clinical practice [25].

Another limitation of the study is the fact that there are no separate groups for comparison. This would be impractical and unethical towards the group with no intervention, who cannot be evaluated with simulation sessions provided training has not been performed. Although the current group studied presents homogeneity, the participants have the same clinical experience, but they did not have any formal training in airway, lung, or cardiac ultrasound before. Still, one reason for bias could be that some of the participants may have participated as observers during the ultrasound evaluation of critical patients in the intensive care unit or in the operating room.

The teaching of the correct placement of the endotracheal tube was performed only in the theoretical part, the mannequin not being able to generate normal airway images, and the evaluation of this aspect was performed only in the theoretical assessment.

## 5. Conclusions

Our proposed curriculum, based on a simple ABC POCUS protocol with an Airway, Breathing, and Circulation approach, is useful in teaching ultrasound basics regarding airway, lung, and cardiac examination using high-fidelity simulation training to anesthesia and intensive care residents. Because airway evaluation was not supported by the simulator, practical skills regarding airway evaluation were not taught and were not evaluated. This represents an area for future research. Most of the anesthesia and intensive care trainees with no expertise in ultrasound have obtained above-average overall scores in the practical examination, with more than 50% obtaining scores in the “good” or “very good” categories. For the theoretical knowledge, 100% of participants passed the 70-score mark in the post-course MCQ test.

The utility of the protocol was not assessed in clinical practice, and the application of the ABC POCUS protocol in critical patients remains an area of interest and future research. Further research is needed to establish the utility of SBME in POCUS and the transition to clinical applications and strategies for objective evaluation of competency in POCUS in critical patients.

## Figures and Tables

**Figure 1 diagnostics-14-00173-f001:**
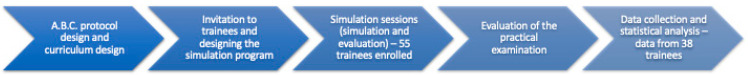
Flow diagram with the study timeline and number of trainees recruited, excluded, and analyzed.

**Figure 2 diagnostics-14-00173-f002:**
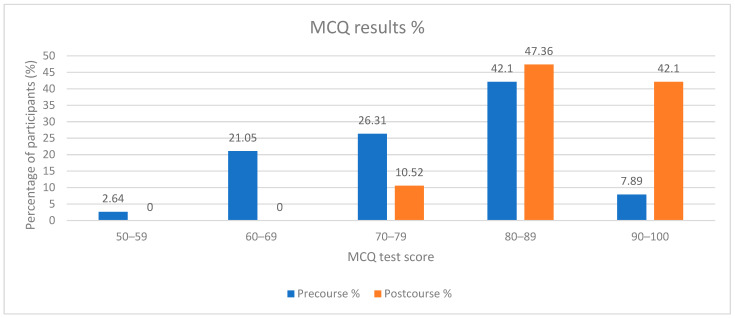
Score results for the pre- and post-course test.

**Figure 3 diagnostics-14-00173-f003:**
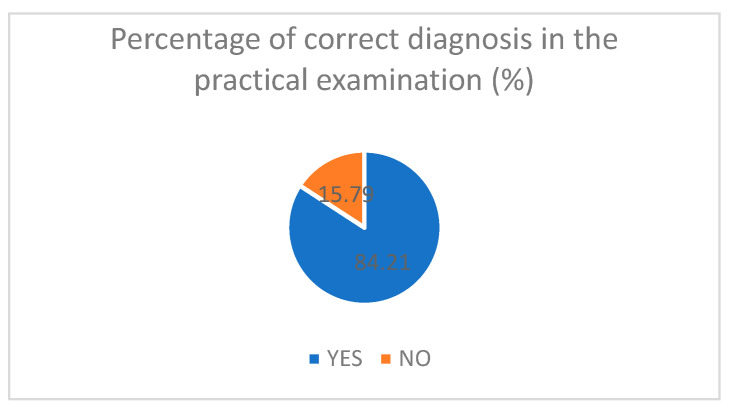
Percentage of correct diagnosis in the practical assessment.

**Figure 4 diagnostics-14-00173-f004:**
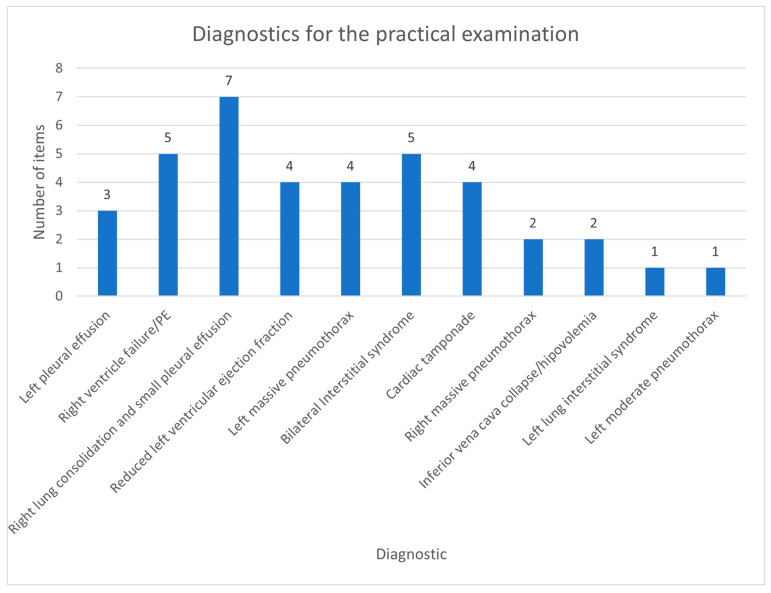
Diagnostics for the practical assessment.

**Figure 5 diagnostics-14-00173-f005:**
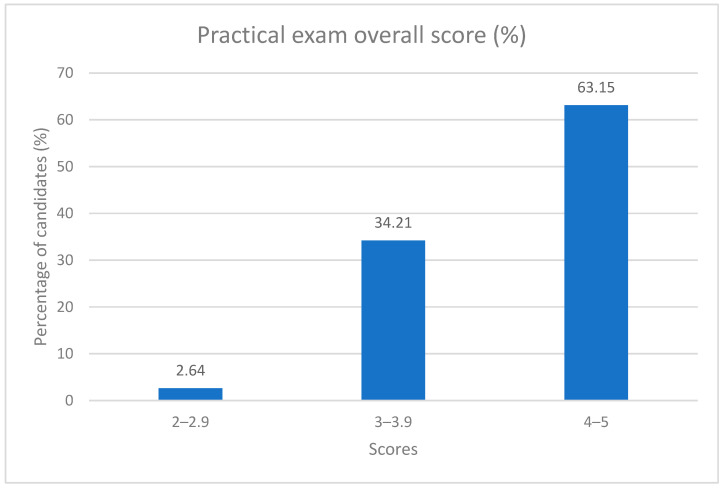
Practical examination overall score (%).

**Table 1 diagnostics-14-00173-t001:** Score test descriptive data.

	N	Mean	SD	SE	Coefficient of Variation
MCQ pre-course	38	77.684	9.804	1.590	0.126
MCQ post-course	38	86.895	5.876	0.953	0.068

**Table 2 diagnostics-14-00173-t002:** Paired Samples *t*-test.

Measure 1	Measure 2	t	df	*p*
MCQ pre-course	MCQ post-course	−7.109	37	<0.001

**Table 3 diagnostics-14-00173-t003:** Interrater agreement for the final practical examination scores.

Ratings	Weighted Kappa
Average kappa	0.471
Ev 1 *–Ev 2 *	0.398
Ev 1 *–Ev 3 *	0.392
Ev 2 *–Ev 3 *	0.622

* Ev 1 = evaluator nr. 1; Ev 2 = evaluator nr. 2; Ev 3 = evaluator nr.3.

**Table 4 diagnostics-14-00173-t004:** Average interrater agreement for separate evaluation criteria.

Criteria nr.	Average Weighted Kappa
1	0.142
2	0.077
3	0.344
4	0.373
5	0.637
6	0.647
7	0.041

## Data Availability

All data are contained within the article.

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
