# Peer review of "Efficiency of Simulation-Based Learning Using an ABC POCUS Protocol on a High-Fidelity Simulator"

_diagnostics, 2024, doi:10.3390/diagnostics14020173_

Round 1

Reviewer 1 Report

Comments and Suggestions for Authors

Would review for minor error and grammar corrections. Use of POCUS and point of care ultrasound interchanged. After acronym defined, ok to use for remainder of paper.

The introduction is a little lengthy with some redundancy. 

Under materials and methods-- ethnical vs ethical.

Figure 2- post test (typo)

Lots of figures and tables. Table 4 is too long. Would find a way to summarize in the text and add that maybe as a supplemental document for reference? Table 1 could also be placed in the supplemental material.

Author Response

I corrected all the typos, use of POCUS acronym after defining it, corrected the figure typo and moved Table 1 and Table 4 separately in the supplemental materials.

Reviewer 2 Report

Comments and Suggestions for Authors

The introduction is well-reasoned and does a good job of outlining why a simulation based curriculum for POCUS would be needed.

Only 69% of the residents completed the entire curriculum which seems low for a curriculum that they volunteered to participate in.  I did not find any evidence that the evaluation criteria were validated or came from a validated scoring system.  

The title of the manuscript is ABC of POCUS but the mannequin did not support learning and evaluation of airway ultrasound skills.  This means that 1/3 of the intended curriculum could not be taught.  Might be better to move this into the conclusions as an area for future curriculum with a mannequin that supports this type of simulation.  Lastly, there was no practical assessment of whether residents used these skills on patients and if simulation based experience allowed them to be proficient with actual patients.  It would be relatively simple to have anesthesia and intensive care residents perform quick evaluations of patients undergoing elective procedures to demonstrate that their technique learned in the simulation lab allowed them to complete an assessment in X amount of time versus residents who had not undergone training.  This type of evidence would greatly increase the enthusiasm for incorporating a new curriculum.

Author Response

I added in the conclusion the importance of future research in the utility and applicability of this protocol in clinical practice and the fact that we were not able to teach and evaluate the practical aspect of airway evaluation with future research and curriculum development needed. I added Table 1 and Table 4 as separate documents that can be included in the supplemental materials.